# Energy Expenditure in Critically Ill Patients with Aneurysmal Subarachnoid Hemorrhage, Intracerebral Hemorrhage, and Traumatic Brain Injury—A Prospective Observational Study

**DOI:** 10.3390/nu16203448

**Published:** 2024-10-11

**Authors:** Geraldine de Heer, Anna Leonie Doliwa, Pascal Hilbert, Marlene Fischer, Patrick Czorlich, Nils Schweingruber, Stefan Kluge, Christoph Burdelski, Jörn Grensemann

**Affiliations:** 1Department of Intensive Care Medicine, University Medical Center Hamburg-Eppendorf, Martinistraße 52, 20246 Hamburg, Germany; 2Department of Neurosurgery, University Medical Center Hamburg-Eppendorf, Martinistraße 52, 20246 Hamburg, Germany; 3Department of Neurology, University Medical Center Hamburg-Eppendorf, Martinistraße 52, 20246 Hamburg, Germany

**Keywords:** nutrition, subarachnoid hemorrhage, intracerebral hemorrhage, traumatic brain injury, neurocritical care, indirect calorimetry, energy expenditure

## Abstract

Background: Energy expenditure (EE) in patients with aneurysmal subarachnoid hemorrhage (SAH) may differ from other intracranial pathologies, such as intracerebral hemorrhage (ICH) or traumatic brain injury (TBI), due to an activation of the sympathetic nervous system. Indirect calorimetry (IC) is recommended, but is not always available. We study EE, catabolism, and metabolic stress in patients with SAH, TBI, ICH, and sepsis as controls. Methods: A prospective observational study was conducted in the intensive care units of the University Medical Center Hamburg-Eppendorf, Germany. IC was used to measure EE on days 2–3, 5–7, and 10–15 post-admission. Urinary catecholamines, metabolites, and urine urea were also measured. Statistical analysis included *t*-tests, Chi-square tests, and generalized mixed models. Results: We included 110 patients—43 SAH patients (13 with the surgical securing of the aneurysm and 30 with coil embolization of the aneurysm), 22 TBI patients, 23 ICH patients, and 22 controls. The generalized linear mixed model analysis for groups and timepoints including age, height, and weight as covariates revealed a significantly lower EE at timepoint 1 for ICH versus SAH—interventional (*p* = 0.003) and versus the control (*p* = 0.004), as well as at timepoint 2 for ICH versus SAH—interventional (*p* = 0.002) and versus SAH—surgical (*p* = 0.013) with a lower EE in ICH patients. No significant differences between groups were found for EE at the other timepoints, or concerning urine urea and measurements of catecholamines in urine. Conclusions: In patients with SAH, ICH, and TBI, no meaningful differences in EE were detected compared to septic critically ill patients, except for a lower EE in ICH patients in the early phase.

## 1. Background

Providing nutrition to critically ill patients is an essential part of intensive care medicine. Overfeeding and underfeeding have both been linked to a decrease in favorable outcome parameters, especially in mechanically ventilated patients [1]. Indirect calorimetry has been recommended as a method to measure energy expenditure (EE) for the guidance of nutritional requirements; however, this method is not ubiquitously available [2,3]. Instead, predictive formulas may be used, but their bias is high, their precision is low [4], and their use may be associated with a suboptimal outcome; however, thus far, few data are available [5].

Patients suffering from subarachnoid hemorrhage (SAH) may exhibit a hypermetabolic response, presumably due to a dysregulation of the sympathetic nervous system, which leads to sympathetic activation [6,7]. This activation leads to an increase in EE [8] and it has been shown that EE may be underestimated by predictive formulas in patients with SAH [9]. A concurrent underfeeding has been associated with poor outcome in SAH patients [10]. During days 2 and 3 after SAH, EE is higher than after day 5 [11]; however, this may only be true for patients receiving aneurysm clipping as opposed to coil embolization [12].

Besides the adoption of nimodipine, drug therapy for the treatment of SAH is not standardized due to a paucity of data; however, it may influence the sympathetic dysregulation of, e.g., dexamethasone, which may attenuate the postoperative inflammatory response but might not show a benefit after coil embolization [13].

The primary objective of this study was to measure EE using indirect calorimetry in neurocritically ill patients to allow for the optimization of nutritional therapy if indirect calorimetry is unavailable. Critically ill patients suffering from sepsis of any cause served as controls. The secondary objectives were to study the influence of co-medication in SAH patients. We hypothesized that EE does not differ between the groups. Furthermore, we hypothesized that the current co-medication in patients with SAH attenuates the sympathetic response and, in addition, that no difference exists between patients with surgical and interventional aneurysm securing and neurocritically ill patients suffering from intracerebral hemorrhage (ICH) or traumatic brain injury (TBI), as well as from the control group.

## 2. Methods

### 2.1. Ethics

This study was approved by the Ethics Committee of the Hamburg Chamber of Physicians (2021-10519-BO-ff; approval: 18 June 2021; chairman: Prof. Dr. Stahl) and consent was waived due to the observational design. The study was conducted in accordance with the Declaration of Helsinki and adheres to the applicable STROBE guidelines. Awake patients were orally informed and gave verbal consent. In unresponsive patients, IC measurements were obtained and the results were used for routine clinical measurements.

### 2.2. Study Design

We conducted a prospective observational cohort study. Due to the nature of the study, no blinding occurred.

### 2.3. Setting and Participants

The study was conducted at the Department for Intensive Care Medicine of the University Medical Center Hamburg-Eppendorf, Germany, including 12 intensive care units (ICUs) with a total of 140 beds and including a dedicated neurosurgical ICU and a neurological ICU with 12 beds each.

Patients were eligible if they were at least 18 years old and suffered from aneurysmal SAH, ICH, or TBI. SAH patients were grouped according to the received treatment modality—(a) microsurgical aneurysm securing; (b) coil embolization. Patients suffering from sepsis served as the control group. Sepsis was defined according to the SEPSIS-3 definition with an increase in the Sequential Organ Failure Assessment (SOFA) score of 2 points or more [14].

We excluded patients with extracorporeal circulation (i.e., extracorporeal membrane oxygenation or renal replacement therapy); with air leaks from the respiratory system (i.e., chest tube with leakage); ventilated patients with an inspiratory fraction of oxygen > 0.7; awake patients receiving supplemental oxygen; and patients receiving a co-medication with catechol-o-methyl-transferase inhibitors (i.e., entacapone or tolcapone), tricyclic antidepressants (TCAs), or selective norepinephrine reuptake inhibitors (SNRIs). Furthermore, patients who had undergone renal replacement therapy (at the current ICU stay or in their ICU history) were excluded as well.

All patients in our department were screened daily for eligibility and were enrolled sequentially.

### 2.4. Nutritional Therapy

Nutritional therapy was independent of this study and was prescribed according to the current German guideline on the nutritional therapy of critically ill patients [3]. In brevity, enteral nutrition was commenced within 24 h after admission and increased according to individual tolerance, aiming for 75% of the estimated target until days 4 to 7. Depending on individual tolerance, 100% nutritional support was started between day 4 and 7. If necessary, parenteral nutrition was added to achieve nutritional targets. For parenteral amino acids, Aminoplasmal 10% (B. Braun, Melsungen, Germany) was used, and a correction factor of 1.2 was applied to calculate the respective dose of amino acids [15].

For the start of nutritional therapy, patients’ requirements were estimated at 24 kcal/kg/d and were adapted to EE after the first indirect calorimetry measurements. Awake patients with adequate swallowing reflexes received oral nutrition ad libitum. Protein intake in awake patients was estimated according to the recommendations of nutritional therapy in patient care [16].

### 2.5. Drug Therapy and Blood Pressure Targets

According to hospital protocols, SAH patients received nimodipine per os or intravenously, if unable to swallow. A mean arterial blood pressure (MAP) of 80 mmHg or above was targeted after the securing of the aneurysm. In cases of vasospasm, MAP was increased to at least 90 or 100 mmHg. The nimodipine dose was reduced if patients required more than 0.25 µg/kg/min norepinephrine to achieve the desired MAP. In patients with ICH, we aimed for a systolic blood pressure between 110 and 140 mmHg. In patients with TBI, a cerebral perfusion pressure of 60 to 70 mmHg was targeted.

### 2.6. Measurements

Patients received indirect calorimetry with Q-NRG+ (COSMED, Rome, Italy) in the “canopy” or “vent-mode”, as applicable. The system was set up according to the manufacturer’s recommendations. Regular calibrations of the indirect calorimetry monitors were performed according to the manufacturer’s specifications, including calibrations of the flow meter and gas probes. During measurements, the system was allowed to stabilize before recording began. Measurements were obtained on the 2nd or 3rd day (timepoint 1—“acute phase”), between the 5th and 7th day (timepoint 2—“end of acute phase”), and optionally between the 10th and 15th day after admission (timepoint 3—“post-acute phase”). All measurements were averaged over at least 5 min according to the manufacturer’s recommendations and were obtained twice on the day of measurement (before noon and in the afternoon with a minimum of six hours in between), and the mean values were used for further calculations. Two hours before the measurements, no interventions were performed, patients were required to rest (e.g., no physiotherapy), and awake patients were measured to determine if the last meal was ingested at least three hours ago.

Furthermore, 24 h urine samples were obtained and total urine nitrogen, metanephrine, and normetanephrine, as well as norepinephrine and epinephrine, were measured. Urine was collected protected from light and on hydrochloric acid to ensure the stability of the catecholamines for analysis. Nitrogen balance was calculated as total protein intake/6.25 − (total urine nitrogen + 4 g) [17].

### 2.7. Outcome Parameters

The primary endpoint was the mean EE. The secondary endpoints were the association of EE and catecholamines and the respective metabolites.

### 2.8. Sample Size

At least fifteen patients per group were required to show a 20% difference in EE between groups for a two-sided test, with a standard deviation of 15%, errors of α = 0.05 and β = 0.2 (G*Power version 3.1.9.2, Heinrich Heine University Düsseldorf, Düsseldorf, Germany), and Bonferroni correction for multiple testing (five groups).

### 2.9. Statistics

Statistical analyses were performed using SPSS (version 27, IBM Inc., Armonk, NY, USA). For the analyses, we used *t*-tests, Chi-square tests, and Fisher’s tests, as applicable. Generalized linear mixed models with sequential Bonferroni correction were used for the simultaneous comparison of differences between groups and timepoints for EE, urine catecholamines, and nitrogen balance. Two-tailed *p*-values < 0.05 were regarded as statistically significant.

## 3. Results

From 1 June 2021 to 28 February 2023, we included a total of 110 patients. An overview of the patient characteristics is depicted in Table 1.

An overview of mean energy expenditure is given in Figure 1 and Table 2. The generalized linear mixed model analysis for groups and timepoints including age, height, and weight as covariates revealed significant differences at timepoint 1 for ICH versus SAH coil (*p* = 0.003) and versus control (*p* = 0.004), as well as at timepoint 2 for ICH versus SAH coil (*p* = 0.002) and versus SAH clip (*p* = 0.013) with a lower EE in ICH patients. All other comparisons between groups and timepoints within groups were not statistically significant. Concerning nitrogen balance, no significant differences were detected between groups (Table 3). No correlation between catecholamines (Table 4) or their metabolites (Table 5) and EE could be established, neither could any meaningful difference be found between the groups for catecholamines and their respective metabolites.

## 4. Discussion

We measured the EE of patients with SAH compared to patients with TBI, ICH, and a control group consisting of critically ill patients suffering from sepsis. We found no difference in EE between the groups, except for a lower EE in patients with ICH.

In critically ill patients, nutrition is an important factor influencing patients’ outcome with data indicating that both underfeeding and overfeeding lead to an increase in mortality [18]; also, mortality rates may increase in patients with SAH [19] and in neurocritical care patients with pre-existing malnutrition [20]. Recent data show a potential outcome benefit for the indirect-calorimetry-guided delivery of nutrition over predictive formulas [21], which are recommended by current guidelines [2,22]. Unfortunately, indirect calorimetry is not ubiquitously available and many clinicians rely on predictive formulas when prescribing nutrition to their patients [23].

In SAH patients, an activation of the sympathetic nerve system has been described, leading to an increase in catecholamine liberation [6,24]. Presumably, in turn, this leads to a hypermetabolic state with an elevated EE, as patients with SAH and endovascular therapy showed a higher EE than patients with acute cerebral infarction [25]. However, data are not consistent with another study, which found a hypermetabolic state in patients with surgical aneurysm securing as opposed to an endovascular approach [12]. For both norepinephrine and epinephrine, an increase in VO_2_ has been shown in the hypermetabolic state. In a study including healthy volunteers, an infusion of 0.2 µg/kg/min norepinephrine increased VO_2_ approximately 20% over baseline [26]. Interestingly, VO_2_ remains elevated even after stopping the catecholamine infusion [27]. These findings support the theory of an increased EE by catecholamine liberation in patients with SAH. The reasons for the activation of the sympathetic nervous system in SAH patients are not fully understood, but neuroinflammation may play a role. This may be attenuated by dexamethasone, as has been shown in meningitis [28] and TBI [29] patients, although no benefit on survival could be shown in the latter [30]. For patients with SAH, a retrospective study has shown an association of dexamethasone therapy with favorable outcomes, but only in patients with microsurgical aneurysm securing and not endovascular therapy [13]. An ongoing prospective randomized controlled trial will give more insights into the effects of dexamethasone on outcome in SAH patients [31]. As we found no meaningful difference in EE between the SAH treatment groups, nor between the SAH and other groups, we cannot exclude the fact that dexamethasone may have played a decisive role in the attenuation of neuroinflammation and the activation of the sympathetic nervous system in our study. Our findings are in contrast to a previous study in which patients receiving surgical aneurysm securing showed a higher EE in the first days after ictus as compared to endovascular therapy [12]. Different treatment strategies for patients who have undergone surgery, such as temperature management, stress avoidance, co-medication, etc., may explain the differences in EE compared to our study.

We cannot give a conclusive explanation for finding a lower EE in patients with ICH, but this certainly indicates that this group should be regarded separately from other disease entities in neurocritically ill patients when conducting research on EE and nutrition. Further studies are needed to elucidate the reasons for a lower EE in this patient population.

Furthermore, we assessed catecholamines and metabolites between the groups without finding a meaningful difference. Physiologically, norepinephrine is metabolized by the catechol-O-methyl-transferase (COMT) to methoxy-norepinephrine, also known as normetanephrine. Epinephrine is metabolized to metanephrine by COMT. Normetanephrine and metanephrine are excreted renally or are further metabolized by monoaminooxydase (MAO) enzymes. Norepinephrine, given as a vasopressor, is also eliminated by COMT and MAO. In our hospital, all patients with SAH received blood pressure augmentation with norepinephrine, targeting an MAP of 80 mmHg or above. In patients with TBI, a cerebral perfusion pressure of 60 mmHg was targeted, also mandating an MAP of approximately 80 mmHg. The exogeneous application of norepinephrine may explain why no differences were detected in our patient cohorts.

Concerning the nitrogen balance, we did not find any difference between the groups. Previously, it has been shown that surgical aneurysm clipping was associated with a negative nitrogen balance [10] and that catabolism was associated with more severe grades of SAH [32]. To the best of our knowledge, no data are available for nitrogen balance in patients suffering from ICH. As opposed to previous studies, the mean nitrogen balances in the groups were less negative and nearly balanced, which we attribute to our early feeding regime and the suppression of inflammation by dexamethasone. Catabolism has been linked to infectious complications, e.g., in SAH [33], but our sample size was too small to study this aspect.

We chose to compare our neurocritically ill patients to septic patients as controls. While hypermetabolism may occur in the course of sepsis, this does not occur during the acute phase and is typical for the recovery phase [34,35].

In addition to the correct estimation of nutritional requirements in a patient group, other factors need consideration. Clinicians should ensure that differences between prescription and actual delivery may occur; one study found a delivery of only 60% of the prescribed nutrition [19]. While only a little bias may exist between indirect calorimetry and predictive formulas, this does not preclude a high inter-patient variability, thus mandating indirect calorimetry for measurements of EE [9].

Our study has the following limitations. Our sample size was small, but sufficient to detect a 20% difference in EE, which we regarded as clinically meaningful. In the SAH group with surgical aneurysm securing, we enrolled two patients fewer than our sample size calculation required. However, we present the first separate cohorts for ICH and TBI patients in comparison to SAH patients and a control group. Our control group consisted of septic patients who themselves may suffer from hypermetabolism due to increased catecholamine levels [36]. We only assessed catecholamines and their respective metabolites in 24 h collected urine without repetitive serum sampling. We cannot exclude that other factors or variations in nutritional support may have influenced EE. This was a single-center study, which may limit the generalizability of our findings to other settings.

## 5. Conclusions

In our prospective evaluation of EE in patients with SAH, ICH, and TBI, no meaningful differences were detected compared to septic critically ill patients serving as controls, except for a lower EE in ICH patients in the early phase.

## Figures and Tables

**Figure 1 nutrients-16-03448-f001:**
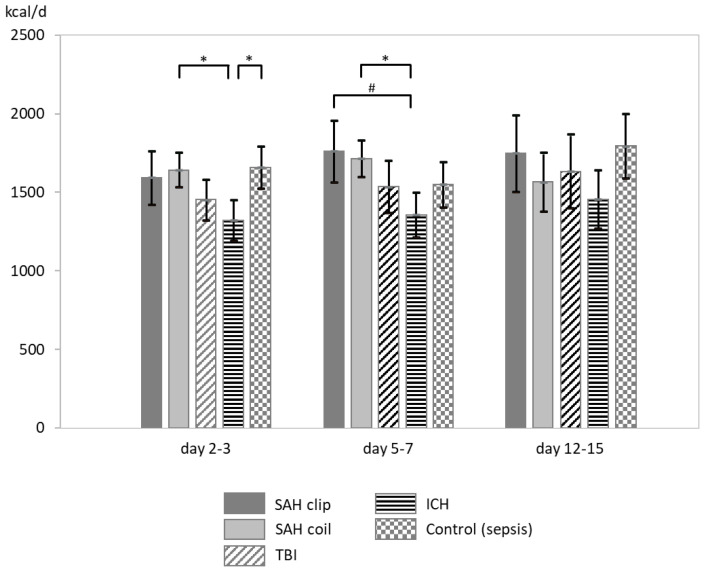
Mean energy expenditure. SAH: subarachnoid hemorrhage; clip: refers to microsurgical aneurysmal securing; coil: refers to aneurysm coil embolization; ICH: intracerebral hemorrhage; TBI: traumatic brain injury. Error bars indicate 95% confidence intervals. * *p* < 0.01; # *p* < 0.05.

**Table 1 nutrients-16-03448-t001:** Patient characteristics.

Parameter	SAH Clip (n = 13)	SAH Coil (n = 30)	TBI (n = 22)	ICH (n = 23)	Control (n = 22)
Age (years)	57 ± 8	57 ± 14	57 ± 18	59 ± 14	67 ± 13
Sex male	n = 2 (15.4%)	n = 10 (33.3%)	n = 14 (63.6%)	n = 12 (63.2%)	n = 12 (50%)
Height (cm)	170 ± 5	172 ± 7	174 ± 8	176 ± 9	171 ± 9
Weight (kg)	72.0 ± 14.1	79.7 ± 13.0	78.5 ± 13.8	82.0 ± 29.5	78.1 ± 21.0
IBW (kg)	62.0 ± 5.6	64.8 ± 8.4	67.4 ± 8.9	70.0 ± 10.2	65.0 ± 9.8
ABW (kg)	66.0 ± 8.2	70.7 ± 8.7	71.9 ± 9.1	74.5 ± 16.2	70.2 ± 10.8
APACHE II	17 ± 8	20 ± 13	21 ± 7	28 ± 11	52 ± 7
SAPS II	29 ± 7	32 ± 11	33 ± 9	34 ± 9	43 ± 6
SOFA	7 ± 4	6 ± 4	8 ± 4	9 ± 3	11 ± 3
Hunt&Hess					
1°	4	8			
2°	2	5	n/a	n/a	n/a
3°	2	5			
4°	1	4			
5°	4	8			
Fisher					
1°	1	5			
2°	1	3	n/a	n/a	n/a
3°	6	7			
4°	5	15			
WFNS					
1°	5	9			
2°	2	4	n/a	n/a	n/a
3°	1	4			
4°	2	5			
5°	3	8			

SAH: subarachnoid hemorrhage; clip: refers to microsurgical aneurysmal securing; coil: refers to aneurysm coil embolization; ICH: intracerebral hemorrhage; TBI: traumatic brain injury; IBW: ideal body weight; ABW: adjusted body weight; APACHE: Acute Physiology And Chronic Health Evaluation, SAPS: Simplified Acute Physiology Score; SOFA: Sequential Organ Failure Assessment; WFNS: World Federation of Neurosurgical Societies; n/a: not applicable. Data are given as mean ± standard deviation or numbers and percentages, as applicable.

**Table 2 nutrients-16-03448-t002:** Mean energy expenditure.

Group	Timepoint 1 (Day 2–3)	Timepoint 2 (Day 5–7)	Timepoint 3 (Day 12–15)
SAH clip	1591 (1420; 1762)	1759 (1563; 1955) ^#^	1746 (1501; 1990)
SAH coil	1641 (1530; 1753) *	1713 (1597; 1829) *	1564 (1377; 1751)
TBI	1451 (1321; 1581)	1534 (1369; 1699)	1632 (1396; 1868)
ICH	1319 (1190; 1448)	1357 (1214; 1499)	1454 (1270; 1638)
Control	1655 (1522; 1789) *	1547 (1403; 1690)	1792 (1587; 1997)

Energy expenditure is given in kcal/d (mean and 95% confidence intervals in brackets); SAH: subarachnoid hemorrhage; clip: refers to microsurgical aneurysmal securing; coil: refers to aneurysm coil embolization; ICH: intracerebral hemorrhage; TBI: traumatic brain injury. * *p* < 0.01 vs. ICH; ^#^
*p* < 0.05 vs. ICH.

**Table 3 nutrients-16-03448-t003:** Nitrogen balance.

Group	Timepoint 1 (Day 2–3)	Timepoint 2 (Day 5–7)	Timepoint 3 (Day 12–15)
SAH clip	−0.6 (−3.0; 1.7)	1.6 (−1.3; 4.5)	0.4 (−3.6; 4.3)
SAH coil	0.3 (−1.3; 1.8)	0.3 (−1.3; 1.9)	2.2 (−0.6; 5.0)
TBI	−2.2 (−4.0; −0.4)	−3.2 (−5.6; −0.9)	−2.7 (−6.2; 0.8)
ICH	−1.4 (−3.1; 0.4)	0.2 (−1.7; 2.2)	0.3 (−2.2; 2.8)
Control	1.4 (−0.5; 3.2)	0.7 (−1.3; 2.7)	1.2 (−2.2; 4.7)

Nitrogen balance is given in g/d (mean and 95% confidence intervals in brackets); SAH: subarachnoid hemorrhage; clip: refers to microsurgical aneurysmal securing; coil: refers to aneurysm coil embolization; ICH: intracerebral hemorrhage; TBI: traumatic brain injury.

**Table 4 nutrients-16-03448-t004:** Catecholamine doses.

Group		Timepoint 1 (Day 2–3)	Timepoint 2 (Day 5–7)	Timepoint 3 (Day 12–15)
SAH clip	NE:	0.178 (−0.113; 0.468)	0.065 (−0.268; 0.398)	0.005 (−0.412; 0.422)
	E:	0	0	0
	Dbt:	0.202 (0.107; 0.297)	0	0
SAH coil	NE:	0.452 (0.262; 0.642)	0.145 (−0.052; 0.342)	0.154 (−0.036; 0.569)
	E:	0	0	0
	Dbt:	0	0	0
ICH	NE:	0.114 (−0.107; 0.336)	0.152 (−0.130; 0.434)	0
	E:	0	0	0
	Dbt:	0	0	0
TBI	NE:	0.047 (−0.170; 0.264)	0.030 (−0.203; 0.264)	0.047 (−0.249; 0.344)
	E:	0	0	0
	Dbt:	0	0	0
Control	NE:	0.139 (−0.083; 0.360)	0.082 (−0.163; 0.326)	0.078 (−0.268; 0.423)
	E:	0	0	0
	Dbt:	0	0	0

Doses are given in µg/kg/min averaged over one day (mean and 95% confidence intervals in brackets); NE: norepinephrine; E: epinephrine; Dbt: dobutamine; SAH: subarachnoid hemorrhage; clip: refers to microsurgical aneurysmal exclusion; coil: refers to aneurysm coil embolization; ICH: intracerebral hemorrhage; TBI: traumatic brain injury.

**Table 5 nutrients-16-03448-t005:** Urine catecholamine and metabolite measurements.

Group		Timepoint 1 (Day 2–3)	Timepoint 2 (Day 5–7)	Timepoint 3 (Day 12–15)
SAH clip	NE:	8.833 (3.202; 14.465)	10.103 (4.705; 15.501)	13.255 (3.034; 23.475)
	E:	5.271 (0.059; 0.483)	0.983 (−0.487; 2.453)	0.492 (0.236; 0.748)
	NorM:	3.944 (1.090; 6.798)	3.892 (2.116; 5.668)	5.978 (3.827; 8.130)
	Meta:	0.878 (0.373; 1.384)	0.750 (0.256; 1.244)	0.887 (0.361; 1.412)
SAH coil	NE:	23.192 (4.245; 42.139)	10.155 (−2.902; 23.212)	5.202 (−0.748; 11.152)
	E:	0.503 (−0.639; 1.645)	0.370 (0.038; 0.702)	0.370 (−0.079; 0.819)
	NorM:	15.497 (0.775; 31.119)	4.849 (−0.936; 10.634)	1.807 (1.019; 2.596)
	Meta:	0.898 (0.072; 1.723)	0.352 (−0.045; 0.749)	0.617 (0.318; 0.916)
ICH	NE:	2.591 (0.439; 4.742)	2.825 (−0.922; 6.572)	3.270 (−1.045; 7.586)
	E:	0.159 (−0.020; 0.337)	0.279 (0.045; 0.514)	0.268 (−0.023; 0.558)
	NorM:	2.059(−0.148; 4.266)	1.878 (0.438; 3.318)	2.053 (0.828; 3.277)
	Meta:	0.904 (−0.029; 1.837)	1.387 (0.238; 2.536)	0.532 (0.175; 0.888)
TBI	NE:	8.623 (−1.728; 18.974)	8.806 (−6.190; 23.802)	21.527 (−33.863; 76.918)
	E:	0.125 (0.026; 0.224)	0.161 (−0.010; 0.331)	0.318 (0.090; 0.546)
	NorM:	9.079 (−4.794; 22.951)	8.273 (−6.968; 23.515)	1.328 (0.640; 2.015)
	Meta:	0.476 (0.115; 0.838)	0.445 (−0.176; 1.065)	0.647 (0.127; 1.166)
Control	NE:	3.338 (0.291; 6.384)	1.131 (0.591; 1.671)	1.278 (0.208; 2.349)
	E:	0.157 (−0.039; 0.354)	0.126 (0.064; 0.187)	0.238 (−0.161; 0.636)
	NorM:	8.213 (−5.296; 21.722)	2.223 (−0.567; 5.012)	1.947 (−2.313; 6.207)
	Meta:	0.845 (−0.354; 2.043)	1.136 (−0.305; 2.577)	0.614 (0.096; 1.132)

Doses are given in µg/kg (mean and 95% confidence intervals in brackets; NE: norepinephrine; E: epinephrine; NorM: normetanephrine; Meta: metanephrine; SAH: subarachnoid hemorrhage; clip: refers to microsurgical aneurysmal exclusion; coil: refers to aneurysm coil embolization; ICH: intracerebral hemorrhage; TBI: traumatic brain injury.

## Data Availability

The data are available from the corresponding author upon reasonable request due to privacy concerns.

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
