# Peer review of "Energy Expenditure in Critically Ill Patients with Aneurysmal Subarachnoid Hemorrhage, Intracerebral Hemorrhage, and Traumatic Brain Injury—A Prospective Observational Study"

_nutrients, 2024, doi:10.3390/nu16203448_

Round 1

Reviewer 1 Report

Comments and Suggestions for Authors

Although the study is prospective, it has a small sample size as the authors indicate in the Limitations, but they should explain in more detail how they corrected for variations in nutrition in each patient so that these did not affect the indirect calorimetry measurements.

On the other hand, an explanatory and comparative graph of the different energy expenditures is missing (perhaps better than a table). Authors need to design a graphic with calorimetric values of each study group and they try to explain how minimize the different nutrition bias among patients.

Author Response

Although the study is prospective, it has a small sample size as the authors indicate in the Limitations, but they should explain in more detail how they corrected for variations in nutrition in each patient so that these did not affect the indirect calorimetry measurements.

> Thank you for this comment. With indirect calorimetry, we measured the total energy expenditure (TEE). Nevertheless, we required patients to rest for two hours before measurements (i.e. no physiotherapy, no transport to diagnostics as CT scan, etc.) and awake patients to remain nihil per os as recommended to obtain resting energy expenditure (REE). We added this information to the manuscript. In critically ill patients, virtually no difference exists between REE and TEE. The TEE is estimated to be only 0-7% higher than the REE (Kreymann et al. Ger Med Sci 2009), with the difference being due to the physical activity level (which we excluded by the resting period) and the dietary induced thermogenesis, that increases energy expenditure, as well. Due to the little difference between REE and TEE, we deemed further correction for variations in nutrition not necessary, particularly because we aimed to detect a 20% difference in energy expenditure, that we regarded as clinically meaningful.

On the other hand, an explanatory and comparative graph of the different energy expenditures is missing (perhaps better than a table). Authors need to design a graphic with calorimetric values of each study group and they try to explain how minimize the different nutrition bias among patients.

> Thank you for this valuable suggestion. We added a figure to the manuscript, graphically depicting the calorimetric values of the studied groups. Concerning the different nutrition bias, please see our above comment.

We thank the reviewer for the helpful comments and suggestions helping us to improve our manuscript. We hope that our manuscript has now sufficiently improved to be acceptable for publication.

Reviewer 2 Report

Comments and Suggestions for Authors

Clearly state the primary objective in a straightforward manner: "The primary objective of this study is ....".

Include any secondary objectives, like the impact of current co-medication on sympathetic nervous system activation?

Clearly state any hypotheses: "We hypothesize that ....".

Explain the rationale for including a control group of septic patients. Discuss the potential limitations of using septic patients as controls, as they might also experience hypermetabolism, potentially affecting the comparative analysis.

Expand on the study design section by including details about the observational nature (e.g., cohort, cross-sectional) and the rationale for choosing this design.

Clearly outline the recruitment process for participants, including how they were identified and selected. Specify any consent procedures, even if consent was waived, to provide clarity on ethical considerations.

Provide more detailed descriptions of the data collection process, particularly concerning the use of indirect calorimetry. Include information on the calibration and maintenance of equipment, and the specific protocols followed during measurements to ensure consistency.

Describe the standard operating procedures for collecting urine samples and conducting indirect calorimetry, including time of day, equipment settings, and environmental conditions during data collection.

Mention any blinding procedures or strategies used to minimize bias, both in data collection and analysis.

Include a more detailed explanation of why specific statistical tests were chosen.

Acknowledge the small sample size.

Note that catecholamine measurements were only done using 24-hour urine samples, which might not capture acute fluctuations or provide as detailed information as serum sampling.

Mention that the study was conducted at a single center, which might limit the applicability of the findings to other settings with different patient populations or care practices.

Discuss any potential confounding factors that were not controlled for, such as variations in nutritional support or other treatments that might influence energy expenditure.

Emphasize the finding of lower EE in ICH patients during the early phase and discuss its clinical significance. Explain why this particular finding is important in the context of neurocritical care and nutritional management.

The discussion mentions other studies related to SAH and energy expenditure. It would be beneficial to compare and contrast these studies more explicitly with the current study's findings.

Include graphs wherever possible. For example, make a line graph out of the data in Table 2, that can illustrate the trends in energy expenditure across different timepoints (days 2-3, 5-7, and 12-15) for each patient group. Other tables can also be convert into graphs (but keep both).

Author Response

Clearly state the primary objective in a straightforward manner: "The primary objective of this study is ....".

Include any secondary objectives, like the impact of current co-medication on sympathetic nervous system activation?

Clearly state any hypotheses: "We hypothesize that ....".

> Thank you for these suggestions. We modified the manuscript as you recommended.

Explain the rationale for including a control group of septic patients. Discuss the potential limitations of using septic patients as controls, as they might also experience hypermetabolism, potentially affecting the comparative analysis.

> Thank you for this remark. We intentionally chose septic as controls, because of the high prevalence of this group in intensive care. Interestingly, hypermetabolism typically occurs in the recovery phase of sepsis, while energy expenditure is not higher in the acute phase. We added a paragraph to the discussion, including two new references on this topic.

Expand on the study design section by including details about the observational nature (e.g., cohort, cross-sectional) and the rationale for choosing this design.

> Our study was a prospective cohort study. The rationale for choosing this design was to be able to follow patients’ disease trajectories taking into account potentially changing energy expenditures. We added the information on cohort study to the manuscript.

Clearly outline the recruitment process for participants, including how they were identified and selected. Specify any consent procedures, even if consent was waived, to provide clarity on ethical considerations.

> We amended the ethics section according to your recommendation.

Provide more detailed descriptions of the data collection process, particularly concerning the use of indirect calorimetry. Include information on the calibration and maintenance of equipment, and the specific protocols followed during measurements to ensure consistency.

> The QNRG+ monitors were set up according to the manufacturer’s recommendations. Regular calibrations were performed as specified by the manufacturer, including calibration of the flow-meter by injecting a specified volume of air as well as calibration of the gas probes with a dedicated test gas mixture provided by the manufacturer. The monitors were maintained at regular intervals as mandated by by-laws on medical equipment and manufacturer’s specifications. We further information to the manuscript.

Describe the standard operating procedures for collecting urine samples and conducting indirect calorimetry, including time of day, equipment settings, and environmental conditions during data collection.

> Urine samples were collected in 5L urine-bags protected from light with a non-transparent black plastic bag. Before collection, 20ml of 6 M hydrochloric acid were given into the urine bag to ensure sufficient acidification of the collected urine. Urine was collected from 6 AM to 6 AM the following day. Indirect calorimetry was performed at noon and in the afternoon with at least 6 hours in between. Before indirect calorimetry, patients were required to rest (no physiotherapy, etc.) and awake patients remained nihil per os. As measurements were obtained in a clinical setting, all measurements were conducted under ATPS conditions (ambient temperature, pressure, saturated).

Mention any blinding procedures or strategies used to minimize bias, both in data collection and analysis.

> Due to the nature of the study, no blinding occurred. We added this information to the manuscript.

Include a more detailed explanation of why specific statistical tests were chosen.

> We added an explanation that Generalized Mixed Models allow for the simultaneous comparison of values between time points and groups.

Acknowledge the small sample size.

> We acknowledge the sample size in our limitations. Nevertheless, we calculated a priori that a sample size of 15 per group would be sufficient to detect a 20% difference in EE. We now include a sentence in the limitations section that the SAH-clip group included fewer patients than our sample size calculation indicated.

Note that catecholamine measurements were only done using 24-hour urine samples, which might not capture acute fluctuations or provide as detailed information as serum sampling.

> This is correct. We intentionally obtained 24h urine samples to have an average over one day as we deemed serum sampling with acute fluctuations as not meaningful.

Mention that the study was conducted at a single center, which might limit the applicability of the findings to other settings with different patient populations or care practices.

> Thank you for this remark. We added this to the limitations section.

Discuss any potential confounding factors that were not controlled for, such as variations in nutritional support or other treatments that might influence energy expenditure.

> Thank you for this remark. We added this to the limitations section.

Emphasize the finding of lower EE in ICH patients during the early phase and discuss its clinical significance. Explain why this particular finding is important in the context of neurocritical care and nutritional management.

> This is a very important point that we did not discuss sufficiently, so far. Unfortunately, we cannot give a conclusive explanation for this finding. However, this indicates that ICH patients present a different disease entity within neurocritically ill patients that needs to be regarded separately from other patient populations. We added a paragraph in the discussion.

The discussion mentions other studies related to SAH and energy expenditure. It would be beneficial to compare and contrast these studies more explicitly with the current study's findings.

> Thank you for this valuable suggestion. We amended the discussion accordingly.

Include graphs wherever possible. For example, make a line graph out of the data in Table 2, that can illustrate the trends in energy expenditure across different timepoints (days 2-3, 5-7, and 12-15) for each patient group. Other tables can also be convert into graphs (but keep both).

> Thank you for this suggestion. We added a figure graphically depicting the data from table 2.

We thank the reviewer for the helpful comments and suggestions helping us to improve our manuscript. We hope that our manuscript has now sufficiently improved to be acceptable for publication.

Round 2

Reviewer 2 Report

Comments and Suggestions for Authors

Thank you for considering my comments. I really appreciate the changes you've made; your article now reads beautifully and has a very professional appearance. Great work!